Health-related issues of people experiencing homelessness in Thailand: a scoping review

http://orcid.org/0000-0002-9537-8510 Wattanapisit Apichai 1 2
http://orcid.org/0000-0001-5901-4640 Muttarat Pannawat 3 pannawat.mu@wu.ac.th
Sottiyotin Tida 3
Puangsri Pavarud 1 2
Aungkawattanapong Nadvadee 4
Wattanapisit Sanhapan 5
Kotepui Manas 6 7
1 School of Medicine, Walailak University , Thasala, Nakhon Si Thammarat , Thailand
2 Walailak University Hospital , Thasala, Nakhon Si Thammarat , Thailand
3 School of Pharmacy, Walailak University , Thasala, Nakhon Si Thammarat , Thailand
4 Department of Pediatrics, King Chulalongkorn Memorial Hospital, Thai Red Cross , Bangkok , Thailand
5 Thasala Hospital , Thasala, Nakhon Si Thammarat , Thailand
6 School of Allied Health Sciences, Walailak University , Thasala, Nakhon Si Thammarat , Thailand
7 Medical Technology Program, Faculty of Science, Nakhon Phanom University , Nakhon Phanom , Thailand
Adegboye Oyelola
Electronic publication date: 2024 Sep 4
Publication date: 2024
Volume: 12
Electronic Location ID: e17884
Received 2024 Apr 11; Accepted 2024 Jul 17
Copyright: © 2024 Wattanapisit et al.
Copyright year: 2024
Copyright holder: Wattanapisit et al.
License: This is an open access article distributed under the terms of the Creative Commons Attribution License, which permits unrestricted use, distribution, reproduction and adaptation in any medium and for any purpose provided that it is properly attributed. For attribution, the original author(s), title, publication source (PeerJ) and either DOI or URL of the article must be cited.
License URL: https://creativecommons.org/licenses/by/4.0/

Keywords: Health, Homelessness, Scoping review, Thailand

Funding: The authors received no funding for this work.

==============================
Introduction

Homelessness is a significant global challenge affecting people worldwide. In Thailand, the health-related issues of people experiencing homelessness have not been a major research focus. This scoping review aims to explore the scope of research on health-related issues among people experiencing homelessness in Thailand.

Methods

Eight databases (ACI, AMED, Embase, PsycINFO, PubMed, Scopus, TCI, and Web of Science) were searched from inception of each database to August 2022. The search terms consisted of terms related to people experiencing homelessness in Thailand. Research articles published in Thai or English were included.

Results

Of 186 articles, 167 were excluded during duplicate removal (n = 45), title and abstract screening (n = 106), seeking for full-text articles (n = 2), and full-text screening (n = 14). Nineteen articles were included for extraction and synthesis. Three topics (health status, causes of homelessness and effects of homelessness on health, healthcare and social services) were summarised. The included studies described mental health (e.g., depression, suicide, alcohol and drug dependence), physical health (e.g., poor self-hygiene, injuries, accidents), and health behaviours (e.g., alcohol drinking, substance use, unsafe sex). Social behaviours and health problems linked to homelessness, and several factors related to health and living conditions (e.g., stigma, discrimination) were reported. People experiencing homelessness had some barriers to access to healthcare and social services (e.g., health insurance, social welfare, financial difficulties).

Conclusions

The majority of studies on health-related issues in people experiencing homelessness in Thailand are descriptive studies. Future studies should focus on the interactions and mechanisms between homelessness and health.

Introduction

Homelessness is a serious violation of human dignity and has become a global issue. It affects people with different backgrounds across the world (United Nations, 2020). According to the United Nations (UN)-Habitat 2015, 1.6 billion people lived with inadequate housing (Kassim et al., 2015). The data in the US showed 580,466 people experienced homelessness in 2020 (70% lived alone and 30% lived with their families) (National Alliance to End Homelessness, 2022). In the European countries, at least 700,000 people experienced homelessness; this number reflected a 70% increase in the past decade (FEANTSA, 2019; World Health Organization & Regional Office for Europe, 2020).

It is known that people experiencing homelessness have higher risks of developing several health problems, compared with other populations (Institute of Medicine (US) Committee on Health Care for Homeless People, 1988). Several health-related issues are highlighted among people experiencing homelessness, including infectious diseases (e.g., HCV, human immunodeficiency virus (HIV)/acquired immunodeficiency syndrome (AIDS)), non-communicable diseases (e.g., diabetes, hypertension), musculoskeletal disorders (e.g., chronic pain), dental health, mental disorders (e.g., depression, anxiety, suicide), childhood trauma, and substance use (e.g., alcohol, methamphetamine) (Mc Conalogue et al., 2021; Oppenheimer, Nurius & Green, 2016; Australian Bureau of Statistics, 2016). These health issues are considered as causes and consequences of homelessness.

More than 4,500 people experiencing homelessness were reported in Thailand (Royal Thai Government, 2022). A previous survey reported that a majority of people experiencing homelessness were males (86%) and 40–59 years old (57%) (Community Organizations Development Institute, 2020). More than half lived in the capital city, Bangkok, and major provinces across the country (Community Organizations Development Institute, 2020). Studies in Thailand reported some health-related issues among people experiencing homelessness. Nicotine addiction (Pitukthanin et al., 2016), violence, physical and sexual abuse (Farley et al., 2004; Techakasem & Kolkijkovin, 2006), and sexual transmitted infection (e.g., HIV/AIDS) (Houysai, Rutchanakul & Kongvattananon, 2015; Narongsakputi, Ratchanakul & Nirattaradol, 2018) were reported as common physical health issues. While psychotic disorders, alcohol dependence, and major depressive episode were identified as major mental health issues (Farley et al., 2004). Thai citizens have been eligible for preventive, curative, and palliative services under the universal health coverage scheme since 2002 (Sumriddetchkajorn et al., 2019; Wattanapisit & Saengow, 2018). However, a survey indicated that people experiencing homelessness had limitations to access to healthcare services (Pitukthanin et al., 2016).

To the best of our knowledge, people experiencing homelessness in Thailand are a minority and less privileged. Studies on health-related issues among this population are not mainstream research in the country. Health-related issues were explored in some studies in the field of social sciences; however, they were not the main research topics. This raises an important question regarding the scope of studies on health-related issues among people experiencing homelessness. This scoping review aims to explore characteristics and scope of studies on health-related issues of people experiencing homelessness in Thailand.

Materials and Methods

This scoping review was conducted and reported following the Preferred Reporting Items for Systematic Reviews and Meta-Analyses extension for scoping reviews (PRISMA-ScR) (Tricco et al., 2018).

Search methods

A systematic search was performed in eight databases, include the Allied and Complementary Medicine Database (AMED), ASEAN Citation Index (ACI), Embase, PsycINFO, PubMed, Scopus, Thai-Journal Citation Index (TCI), and Web of Science. The search included articles from inception of each database to 9th August 2022. The search terms related to people experiencing homelessness in Thailand: (homeless OR homelessness OR houseless OR roofless OR unhoused OR “rough sleeping”) AND (Thailand OR Thai) were adapted to each database’s command language. For the ACI and TCI databases, only the search terms for homelessness (not including Thailand) using the Boolean ‘OR’ were searched due to a limitation of the databases. The Thai words for homelessness (i.e., rai-ban, rae-ron) were searched in a TCI search. Articles found from the databases were transferred to Endnote X9 citation manager (Clarivate, Philadelphia, PA, USA).

Study selection and eligible criteria

After the duplicate removal, two authors (AW and TS) independently screened titles and abstract. Any disagreement was reviewed by another author and resolved through consensus. The authors (AW and TS) reviewed full-text articles for eligibility based on the following inclusion and exclusion criteria. A full-text article was included if: (i) the article related to health-related issues of people experiencing homelessness in Thailand and (ii) the article was published in the English or Thai languages. A full-text article was excluded if: (i) the article was a non-empirical study (e.g., systematic review, scoping review, narrative review), conference abstract, book or book chapter, and (ii) the article focused on homelessness due to criminal justice (e.g., prison).

Data extraction and synthesis

The two authors (AW and TS) extracted information, including, title, author names, publication year, language of publication, study design and data collection, participant and setting, outcome measurement, and results, from the included articles. The information was recorded in the data extraction form (Box 1). The authors synthesised the articles to summarise topics on health-related issues among people experiencing homelessness.

Box 1 Title:	
Author names:	
Publication year:	
Published language:	
Study design and data collection:	
Participant and setting:	
Outcome measurement:	
Results:	

Results

Of 186 articles retrieved from the eight databases, 45 duplicates were removed. One hundred and six articles were excluded during the title and abstract screening. Two full-text articles were not available on journals’ websites, and fourteen articles were excluded after the full-text screening. A total of 19 articles (19 studies) were included for analysis (Fig. 1).

Figure 1 PRISMA flow diagram.

Characteristics of the included studies

Of 19 studies, 13 studies (68.4%) were published in Thai and six studies (31.6%) were in English language. Seventeen studies (89.5%) were published in the past decade (2014–2022), and two studies (10.5%) were published before 2010 (2004 and 2006). Six studies (31.6%) employed a qualitative design, and three studies (15.8%) were cross-sectional studies. Five studies (26.3%) focused on children and adolescents. Nine studies (47.4%) were conducted among staff or officers who work with people experiencing homelessness. Of the nine studies involved officers, five studies included people experiencing homelessness as study participants. One study (5.3%) was conducted in nine countries, including Thailand. The rest of the studies (n = 18, 94.7%) were conducted in Thailand only. Eight studies (42.1%) were conducted in the capital city. A variety of study designs (e.g., cross-sectional study, experimental study, qualitative study, mixed methods study, action research) was conducted. Most studies were descriptive studies. Table 1 shows the summary of the included studies. Three topics were summarised from the included studies.

Table 1 Summary of the included articles (n = 19).

Citation (language of publication)	Study design and data collection	Participants and setting	Outcome measurement	Health-related issues	
Farley et al. (2004) (English)	Study design:
Cross-sectional study
Data collection:
Brief structured interviews using questionnaires: prostitution questionnaire and PTSD checklist	854 people in prostitution in nine countries, 110 respondents from Thailand (two cities)	History of prostitution, violence in prostitution (including current or past homelessness), types of violence, prostitution and pornography, PTSD	57% of Thai respondents experienced current or past homelessness; 39% threatened with a weapon; 56% physically assaulted; 56% raped; 47% sexually abused as a child; 45% pornography made in prostitution; 58% PTSD	
Techakasem & Kolkijkovin (2006) (English)	Study design:
Cross-sectional study
Data collection:
Medical record reviews (for runaways and non-runaways in the mental health service) and semi-structured interviews (for runaways in the juvenile justice system)	Three groups of children and adolescents: 21 runaways in the mental health service in Bangkok, 21 non-runaways in the mental health service in Bangkok, and 21 runaways in the juvenile justice system	Demographic characteristics and psychiatric diagnoses (runaways the mental health service vs. non-runaways the mental health service)
Demographic characteristics (runaways the mental health service vs. runaways in the juvenile justice system)	In the mental health service, runaways were more likely to have the neglect (33.3% vs. 0), sexual abuse (38.1% vs. 0), rejection (42.9% vs. 4.8%), poverty (52.4% vs. 9.5%), and being in custody (38.1% vs. 4.8%), borderline disorder (19.0% vs. 0), depression (33.3% vs. 19.0%), conduct disorder (42.9% vs. 9.5%), and substance abuse (47.6% vs. 0)
Comparing between runaways in different settings, runaways in the juvenile justice system were more likely to have physical abuse (71.4% vs. 23.8%), authoritarian (76.2% vs. 33.3%), and being in custody (95.2% vs. 38.1%)	
Muannadon et al. (2019) (Thai)	Study design:
Qualitative study
Data collection:
Focus group discussions and observation	16 government officers affiliated with nine organisations, four children experiencing homelessness, and two people who worked closely with street children in Udon Thani Province	Current situations and patterns of government support
Suggestions for the care of children experiencing homelessness	Family breakdown and poverty, substance misuse, and attachment to their peers were the reasons to live on the street
Peer relationships, freedom, and the earning potential were the main reasons to continue being homeless
The children experiencing homelessness were at risk of low self- esteem, behavioural problems, substance misuse, STIs, stigma, and discrimination
The suggestions included developing family bonds and warmth to prevent leaving homes and supporting essential needs (e.g., education, health, employability)	
Houysai, Rutchanakul & Kongvattananon (2015) (Thai)	Study design:
Two-group quasi-experimental study (intervention: empowerment programmes to prevent STIs; evaluation: pre- and post-tests)
Data collection:
Questionnaires	28 male adolescent vagrants aged 13–18 years in Bangkok	Pre- and post-test scores of STI preventing behaviours
(e.g., sexual arousal management, sexual behaviours, sexual hygiene, observation of genital abnormalities, sexual risk assessment)	The post-test scores of STI preventing behaviours were significantly improved after the 4-week programme	
Viwatpanich (2015) (English)	Study design:
Mixed methods study
Data collection:
In-depth interviews and structured questionnaires	60 people aged ≥60 years experiencing homeless for 2–5 years in Bangkok	Live story (qualitative data)
Demographic characteristics, causes of homelessness, lifestyle, problems and obstacles, health status, social and health needs, and advantages of
homelessness (quantitative data)	Health problems and disabilities were the most cited causes of permanent homelessness (30%)
Homelessness affected living patterns that related to health, including, cooking, housing, bathing, cleaning, toileting, and relaxing; increased risks of physical and sexual abuse.
63.3% of participants reported health problems before becoming homeless: hearing problems (10%), peptic ulcers (10%), DM (6.7%), and cardiovascular diseases, hypertension, kidney diseases, muscle pain (5%) and 25% of participants had cognitive impairment
Participants rated their health status as healthy (16.7%), acceptable (46.7%), and bad (36.7%); most participants (95%) could independently perform daily activities; 75% and 78.3% had mental stress and depression, respectively while 70% felt happier than the general population
New patterns of health problems, including musculoskeletal pain, tuberculosis, asthma, skin diseases, and injuries and accidents were commonly found; 10% required hearing aids and wheelchairs; good hygienic housing was mentioned as a place for sleep and protecting them from infectious and skin diseases and mosquitoes	
Khongmueang, Ekakun & Nilmoje (2017) (Thai)	Study design:
Research and development (four phases: analysis, design and development, implementation, and evaluation)
Data collection:
Questionnaires, observation, interviews, and focus group discussions	40 children and youth experiencing homelessness (15 participated in the implementation and evaluation phases), two teachers, and five content experts in Ubon Ratchathani Province	Social and educational contexts and educational programme evaluation	Children and youth experiencing homelessness strayed and wandered aimlessly; had family poverty and a lack of close-knitted family; were at risk of unemployment, game addiction, drug dependence, and unsafe living (e.g., natural disasters)	
Awirutworakul et al. (2018) (English)	Study design:
Cross-sectional study
Data collection:
30-min interviews by trained healthcare professionals and social workers	113 Thai-ethnicity individuals (82.3% males, aged 48.6 ± 11.6 years) experiencing homelessness in Bangkok	Prevalence of psychiatric disorders and mental problems using the MINI, DSM-IV, ICD-10
associations between psychiatric comorbidities and severe psychiatric
disorders (psychotic and related disorders)	Prevalence of overall psychiatric disorders = 76.1%; major depressive episode (current, 2 weeks) = 35.7%; psychotic disorders (lifetime) = 31%; suicidality (current, past month) = 29.2%; psychotic disorders (current) = 23%; and alcohol dependence (past 12 months) = 22.1% (data shown only the prevalence ≥ 20%)
Presence of suicidal risks (p = 0.003, OR 4.28, 95% CI [1.52–12.04] and multiple psychiatric disorders (p = 0.001, OR 22.94, 95% CI [6.20–84.98]) were associated with severe psychiatric disorders	
Narongsakputi, Ratchanakul & Nirattaradol (2018) (Thai)	Study design:
Mixed methods study
Data collection:
Questionnaire (the sexual and reproductive health behaviour)	153 adolescents experiencing homelessness aged 13–19 years in Bangkok	Sexual and reproductive health behaviours and problems	Overall sexual and reproductive health behaviours were rated at low to moderate levels; self-observation of abnormal symptoms related to STIs and self-hygiene were rated at a low level
57.28% had more than one sex partner in the past 3 months; 85.44% had sexual activities with non-regular partner at least 1 time/week; 12.62% were pregnant or involved in pregnancy of their partners; 8.74% had experience of abortion; and 90.20% did not receive medical services at healthcare institutions
Problematic behaviours included (i) STIs and abnormalities of the genitalia; (ii) unwanted pregnancy and induced abortion; (iii) limited knowledge of STIs	
Srisung & Srivichaimool (2018) (Thai)	Study design:
Action research
Data collection:
Document reviews and focus group discussions for developing learning curriculum
Questionnaires for assessing the children’s learning abilities before and after the use of the curriculum	24 children experiencing homelessness and six teachers in Chiangrai Province	Learning abilities before and after implementation of the curriculum	All learning domains (i.e., social skills, life skills (e.g., self-care, hygiene), communication and basic mathematic skills, work skills, information and technology skills, overall) were significantly improved after the use of the curriculum.	
Worahan & Rojanaworarit (2018) (Thai)	Study design:
Retrospective analytic study
Data collection:
Medical record reviews	468 males experiencing homelessness who lived in a public shelter and received dental check-ups at a hospital dental department in 2018 in Pathum Thani Province	General health conditions and oral health status	Disabilities (77.1%): mental (65.4%); physical (35%); intellectual (5.3%)), alcohol drinking (42.7%), and smoking (71.1%) were recorded
Oral problems were having less than 20 functional remaining teeth (52.1%); periodontal disease (97.3%); untreated decayed teeth (2.7 teeth/person); missing teeth (12.8 teeth/person); filled teeth (0.1 teeth/person)	
Khutkhong et al. (2020) (Thai)	Study design:
Qualitative study
Data collection:
Document reviews, in-depth interviews, and non-participation observation	17 respondents (five social workers, psychologist, social developers. and academics; and 12 people experiencing homelessness)
in Surat Thani Province	Behaviours and social interactions of people experiencing homelessness
Recommendations for improving quality of life of people experiencing homelessness	Two main groups were missing people and people with mental illnesses
Causes of homelessness: (i) mental illness (e.g., cognitive impairment –unable to go home and contact their families) and (ii) family problems (e.g., family burden due to mental illnesses)
Common health problems: (i) skin diseases (e.g., ringworm, tinea, rashes, ulcers, body odour); (ii) oral health (e.g., dental carries); (iii) respiratory diseases due to infections and pollution; (iv) mental health (e.g., stress, depression, anxiety, suicidal ideation); (v) alcohol drinking; and (vi) self-hygiene (e.g., bathing, clean food, clean sleeping spaces)
Recommendations included providing comprehensive needs (e.g., access to healthcare services, housing, employment, education) and protection of rights and social welfare without discrimination	
Luangsurin, Metiyothin & Wiroonratch (2019) (Thai)	Study design:
Participatory action research
Data collection:
In-depth interviews, observation, and focus group discussions	Officers who work in the Thailand department of social development and human security and organisations that play roles in homelessness management	Risk management model for protection of people experiencing homelessness	Internal risks of the organisations included (i) client (e.g., communicable diseases, behaviours); (ii) practice (e.g., ineffective methods); (iii) human resource (e.g., limited human resources); (iv) finance (e.g., insufficient budget)
External risks of the organisations were (i) social aspects; (ii) community aspects; (iii) legal aspects; and (iv) economic aspects	
Tangtammaruk & Chaiwat (2019) (English)	Study design:
Cohort study
Data collection:
Interviews during weekly visits for 5 months	90 people experiencing homelessness ≤5 years in a government shelter, an NGO shelter, and public places in Bangkok	Happiness and its associated factors, including personal factors, relationship factors, way of living factors (consisted of some health behaviours), and income and economic factors	Two health behaviours were significantly associated with happiness: having snack (small meals) every day (positive effect: 66.70% increase of probability of happiness)
and smoking (negative effect: 51.96% decrease of probability of happiness)
Other health behaviours showed non-significant association with happiness: positive factors (i.e., having sufficient food per day) and negative factors (i.e., having sleep problems, drinking alcohol)	
Wongjongrungruaeng, Naowarat & Kaewyot (2019) (Thai)	Study design:
Cross-sectional study
Data collection:
Medical record reviews using data record forms	261 people experiencing who were admitted to a psychiatric hospital in Bangkok between 2017 and 2018	Demographic characteristics, transferring processes to the hospital, and diagnoses and treatments	53.3% were males and 35.2% had past medical illnesses; 66.7% were transferred to the hospital by policemen/soldiers/government officers and 70.2% were found in public places; common diagnoses were schizophrenia (64.4%), substance used disorders (16.1%), unspecified nonorganic psychosis (7.7%), and bipolar disorder (1.9%); mean length of admission was 53.6 ± 46.5 days, hospital discharge by going home with relatives (44.4%), going to public shelters (30.3%), referring to other hospitals (9.2%)	
Yodkeeree & Laochankham (2020) (Thai)	Study design:
Qualitative study
Data collection:
In-depth interviews	14 participants (seven government officers, two representatives from an NGO, and five people experiencing homelessness living in shelters) in Khon Kaen Province	Assistance and protection of people experiencing homelessness	People experiencing homelessness may not have documents for health insurance and social welfare which were the barriers to assistance; may have a distrust of people, including government officers due to their past history (e.g., crime, infectious diseases); may have the social stigma (e.g., crime, infectious diseases, danger) which caused physical and mental difficulties
Drinking alcohol in public spaces was a problem that caused problematic alcohol use (e.g., alcohol dependence, mental problems, physical assaults)	
Piamsap & Laochankham (2020) (Thai)	Study design:
Qualitative study
Data collection:
Semi-structured in-depth Interviews	Eight government and NGO servants, including, social workers, policemen, social development worker, community development worker, and coordinator of a homeless network in the urban area of Khon Kaen Province	The management of relevant agencies for people experiencing homelessness with mental illnesses according to the mental health act	Several agencies involved in the processes of the care of people experiencing homelessness with mental illnesses, including pre-, peri-, and post-hospitalisation; both active and passive policies were advocated in the area
Some challenges were addressed: (i) human resources (i.e., lack of knowledge of mental illness among non-healthcare workers, unclear roles, lack of specialist in mental illnesses among people experiencing homelessness) and (ii) management (i.e., lack of sustainability, lack of communication among relevant agencies)	
Thienwiwatnukul, Ngamthipwatthana & Phattharayuttawat (2020) (English)	Study design:
Mixed methods study
Data collection:
Structured interviews (for diagnosing psychiatric disorders) and semi-structured interviews (for healthcare-seeking behaviours)	116 people aged ≥18 years experiencing homeless residing in two shelters in Bangkok	Prevalence of psychiatric disorders using the MINI, DSM-IV, ICD-10
Healthcare-seeking behaviours	At least one psychiatric disorder (lifetime prevalence) = 61.2%; alcohol dependence (past 12 months) = 21.6%; substance dependence (past 12 months) = 9.5%; suicidal risk = 31.0%
Having ≥1 psychiatric disorder was associated (higher risks) with gender (male), shelter place (public shelter), place of living before residing in a shelter (public space) and criminal activity involvement
Barriers to utilising healthcare services: payment and health coverage (32.76%); transportation and travelling (21.56%); feeling of receiving sub-standard care (12.1%); healthcare providers’ disrespectfulness and discrimination (5.17%)	
Tuancharoensri & Nunnuan (2020) (Thai)	Study design:
Qualitative study
Data collection:
Structured interview	16 government officers (five executives and 11 social developers) from five regions across the country	Roles and challenges of the government agency for preparing people with psychiatric disorders before returning to the society	The government protection centre had crucial roles to (i) categorize, analyse, support people with psychiatric disorders before returning to the society; (ii) prepare the family; (iii) follow up with home visits
Challenges were (i) policy aspects (non-context specific indicators); (ii) management aspects (insufficient human workforce); (iii) personnel aspects (lack of expertise); (iv) operational aspects (limited information from primary sources); and (v) collaborative aspects (lack of collaboration)	
Sattayawatee, Anuttaranggoon & Promwichai (2021) (Thai)	Study design:
Qualitative study
Data collection:
Interviews	Government officers who involved in the care of people experiencing homeless in Nakhon Sawan Province	Trends of the number of homelessness and roles of the government sector in caring for people experiencing homeless	A decreased number of homelessness was reported (decreased from 200 to 90 in the study cite)
The centre for the protection of the defenceless was the main agency as a screening unit, analysing problems, evaluating target groups, and coordinating referrals to relevant agencies	
Note:

95%CI, 95% confidence interval; DM, diabetes mellitus; DSM-IV, diagnostic and statistical manual, fourth edition; ICD-10, international classification of disease, tenth edition; MINI, mini-International Neuropsychiatric Interview; OR, odds ratio; NGO, non-governmental organisation; PTSD, post-traumatic stress disorder; STI, sexually transmitted infection.

Topic 1: health status

Various health conditions, including mental health, physical health, and health behaviours were reported. One study showed 36.7% of people experiencing homelessness rated their health status as bad, while about half (46.7%) perceived their health status was acceptable (Viwatpanich, 2015).

High rates of depression (78.3%) and mental stress (75%) were reported (Viwatpanich, 2015). Two studies reported more than six in ten had at least one psychiatric disorder (Awirutworakul et al., 2018; Thienwiwatnukul, Ngamthipwatthana & Phattharayuttawat, 2020). Suicidal risks and alcohol dependence were reported at approximately 30% and 20%, respectively (Awirutworakul et al., 2018; Thienwiwatnukul, Ngamthipwatthana & Phattharayuttawat, 2020). About one third of people experiencing homelessness had major depression (35.7%) and psychotic disorders (31%) (Awirutworakul et al., 2018). Severe psychiatric disorders were associated with higher risks of suicide and multiple psychiatric disorders (Awirutworakul et al., 2018).

One study analysed the characteristics of patients admitted to a psychiatric hospital and found that 35.2% had underlying illnesses (Wongjongrungruaeng, Naowarat & Kaewyot, 2019). Among patients admitted to a psychiatric hospital, schizophrenia (64.4%), substance use disorders (16.1%), unspecified nonorganic psychosis (7.7%), and bipolar disorder (1.9%) were common diagnoses (Wongjongrungruaeng, Naowarat & Kaewyot, 2019).

Physical health issues such as musculoskeletal pain, tuberculosis, respiratory diseases, skin diseases, poor self-hygiene, injuries, and accidents were commonly found among older adults experiencing homelessness (Viwatpanich, 2015; Khutkhong et al., 2020). A study in 468 people experiencing homlessness who had dental check-ups found 97.3% had periodontal diseases and 52.1% had insufficient functional teeth (Worahan & Rojanaworarit, 2018).

A study on adolescents experiencing homelessness’ sexual behaviour showed that 57.28% had multiple sex partners and 12.62% experienced pregnancy or involved in pregnancy of their partners (Narongsakputi, Ratchanakul & Nirattaradol, 2018). Other health-related behaviours such as alcohol drinking, substance use, and poor self-hygiene were reported in the studies (Muannadon et al., 2019; Khutkhong et al., 2020).

Topic 2: causes of homelessness and effects of homelessness on health

Social behaviours and experiences linked to homelessness. A cross-sectional study conducted among people in prostitution found that 57% experienced current or past homelessness (Farley et al., 2004). Compared to youth who live with their families, youth who left homes were more likely to child neglect, sexual abuse, rejection, poverty, and being in custody (Techakasem & Kolkijkovin, 2006). Another study found that children and youth experiencing homelessness were more likely to have family poverty and a lack of close-knit family (Khongmueang, Ekakun & Nilmoje, 2017). Reasons for leaving home among children included family problems (such as breakdown and poverty), substance use, and attachment to their peers (Muannadon et al., 2019). A study conducted in older adults found that health problems and disabilities were common causes of permanent homelessness (Viwatpanich, 2015). Cognitive problems and mental illnesses were identified as the main reasons of being homeless in one study (Khutkhong et al., 2020).

Several factors related to health and living conditions were reported among people experiencing homelessness. A study reported that having extra daily meals had a positive effect on happiness, while smoking led to a negative impact on happiness (Tangtammaruk & Chaiwat, 2019). Homelessness affected living patterns related to health (e.g., hygiene, physical abuse, sexual abuse) (Viwatpanich, 2015). People experiencing homelessness who drank alcohol in public spaces were more likely to have alcohol-related problems such as physical assaults (Yodkeeree & Laochankham, 2020). Children and youth experiencing homelessness were at risks of unemployment, game addiction, drug dependence, unsafe living, low self-esteem, behavioural problems, stigma, discrimination, and sexually transmitted infections (Khongmueang, Ekakun & Nilmoje, 2017; Muannadon et al., 2019). Sexual and reproductive health problems, including sexually transmitted infections, unwanted pregnancies and induced abortions, and limited knowledge of sexual diseases, were reported among adolescents experiencing homelessness (Narongsakputi, Ratchanakul & Nirattaradol, 2018).

Topic 3: healthcare and social services

Different issues related to healthcare and social services were reported. The main focuses included barriers to services, management, and interventions.

Barriers to access to healthcare and social services included payment, documents for health insurance and social welfare, transportation, perception of receiving sub-standard care, healthcare providers’ disrespectfulness, discrimination, and social stigma (Thienwiwatnukul, Ngamthipwatthana & Phattharayuttawat, 2020; Yodkeeree & Laochankham, 2020). One study highlighted the challenges of relevant public agencies to provide care for people experiencing homelessness, which included human resources (e.g., non-healthcare workers’ knowledge) and management (e.g., lack of communication among relevant agencies) (Piamsap & Laochankham, 2020).

A study showed the roles of the government sector in caring for people experiencing homelessness in Nakhon Sawan Province, which included screening unit, analysing problems, evaluating target groups, and coordinating referrals to relevant agencies (Sattayawatee, Anuttaranggoon & Promwichai, 2021). To improve the quality of life of people experiencing homelessness, providing comprehensive support (e.g., healthcare services, housing, employment, education) and protection of rights and social welfare without discrimination were recommended (Khutkhong et al., 2020). One study reported policy aspects, personnel aspects, and collaborative aspects as challenges to prepare people with psychiatric disorders before returning to the society (Tuancharoensri & Nunnuan, 2020). Another study identified risk management of orgnisations, including internal (e.g., human resource) and external risks (e.g., legal aspects) for protection of people experiencing homelessness (Luangsurin, Metiyothin & Wiroonratch, 2019).

One study presented the development and effectiveness of a learning curriculum for children experiencing homelessness, and the results showed the improvement of several domains, including life skills (e.g., self-care, hygiene) (Srisung & Srivichaimool, 2018). A study assigned empowerment programmes to prevent sexually transmitted infections and found the significant improvement of the scores of sexually transmitted infection preventing behaviours (Houysai, Rutchanakul & Kongvattananon, 2015).

Discussion

This scoping review identified 19 studies with regard to health-related issues of people experiencing homelessness in Thailand. Three topics (health status, causes of homelessness and effects of homelessness on health, healthcare and social services) were summarised from the included studies. Several studies illustrated mental health, physical health, and health behaviours among people experiencing homelessness. Homelessness affected people’s lives as the bidirectional interactions. Homelessness was both a cause and an effect of health problems. People experiencing homelessness had some barriers to access to healthcare and social services (e.g., health insurance, social welfare, financial difficulties, social discrimination and stigma). A number of studies explored the roles and needs of the government sector to provide care for people experiencing homelessness. Two studies showed the effectiveness of educational and health promoting programmes to improve health of children and youth experiencing homelessness.

Studies in Thailand revealed the connection between mental health and people experiencing homelessness. Psychiatric conditions, including mood disorders, psychosis, alcohol and drug dependence, were reported as important mental health issues. This finding was consistent with studies conducted in other countries (Carrillo Beck et al., 2022; Doran et al., 2018; Onapa et al., 2022). There were limited studies regarding physical health among people experiencing homelessness in Thailand. In this review, the outstanding concerns included accidents, injuries, and health conditions associated with hygiene. Previous literature shared the similar trends of physical health among people experiencing homelessness (Fazel, Geddes & Kushel, 2014). Sexually transmitted infections were found to be a main focus among some studies in this review. The high prevalence and vulnerability of sexually transmitted infections in people experiencing homelessness were also reported in studies from other regions of the world (Gonçalves Barbosa et al., 2023; Segala et al., 2024; Williams & Bryant, 2018). A variety of health-risk behaviours, including risky alcohol use, heavy smoking, and illicit drug use among people experiencing homelessness were commonly reported, which were in line with other studies in Thailand (Padilla et al., 2020; Smith-Grant et al., 2022). In contrast, physical and sexual abuse were uncommonly reported in Thai studies.

Several studies from Thailand demonstrated homelessness as the cause and also the consequence of health-related problems. From social health perspective, family issues (e.g., domestic violence, family breakdowns, unsafe living conditions) are some evidenced causes of being homeless (Karabanow, 2008; van den Bree et al., 2009). Considering neuropsychiatric conditions, mental illnesses and cognitive impairment were reported some causes of homelessness in Thai studies (Khutkhong et al., 2020). Previous evidence also supported this finding (Stone, Dowling & Cameron, 2019). On the other hand, homelessness affects physical, mental, and social health. This was in line with studies in other countries (Bower, Conroy & Perz, 2018; Duke & Searby, 2019; Liu, Chai & Watt, 2020; Nanjo et al., 2020). Most studies conducted in Thailand revealed the negative health conditions among people experiencing homelessness. In contrast, a study described a majority of people experiencing homelessness perceived that they were happier than the general population (Viwatpanich, 2015). However, this result was collected by interviewing the study participants, and it was perspective, not compared to the perception of happiness among the general population.

Most included studies in this scoping review focused on social services (e.g., documents for health insurance and social welfare) rather than healthcare services. The Thai studies emphasised the characteristics of and barriers to social services. Public policies or strategies to prevent homelessness were not found in the included studies. A few studies addressed barriers to healthcare such as health insurance and payment (Yodkeeree & Laochankham, 2020; Thienwiwatnukul, Ngamthipwatthana & Phattharayuttawat, 2020). These issues affected the access to healthcare services. Some studies focused on utilising mental health services among people experiencing homelessness in Thailand (Wongjongrungruaeng, Naowarat & Kaewyot, 2019; Piamsap & Laochankham, 2020). However, studies in Thailand did not focus on the spectrum of healthcare services, including health promotion, disease prevention, treatment, rehabilitation, and palliation (Boerma et al., 2014; Klinjun et al., 2022). The included studies in this review revealed the health status of people experiencing homelessness rather than the healthcare service utilisation. This reflected a lack of studies on the mechanisms of healthcare service utilisation among people experiencing homelessness in Thailand.

There were some strengths of this scoping review. First, the systematic search was performed in eight databases, including both international and domestic databases. The Thai language search was applied to the TCI database. This approach could improve the yields of search results for Thai literature. Second, this scoping review focuses on a broad range of health-issues to explore the existing evidence from different academic fields and identify the gap in knowledge. The research paradigm and values could contribute to a better understanding of the knowledge in this field. Third, the search included all types of study designs. This helped identify the characteristics of studies on people experiencing homelessness conducted in the Thai context. Four limitations were addressed. First, the search terms for the ACI and TCI databases were adapted for the suitability for the databases. Second, this scoping review was unable to summarise the overall quantitative data of studies (e.g., pooled prevalence of psychiatric disorders). Third, some of the included studies presented the perspectives of individuals who looked after people experiencing homelessness rather than those of the people experiencing homelessness themselves. This could introduce bias into this review. Fourth, the quality assessment of individual studies in this review was not conducted. As a scoping review, this process was not compulsory (Peters et al., 2021).

Conclusion

This review sheds light on the scope of health-related issues in people experiencing homelessness in Thailand. Studies in Thailand describe the characteristics of mental health, physical health, and health behaviours among people experiencing homelessness. Social behaviours and health problems have a connection with homelessness. Homelessness affects health and living conditions. Social services are described in a number of studies. However, healthcare services for people experiencing homelessness are not the main focus of studies conducted in Thailand. Most studies included in this scoping review are characterised as descriptive data. The analytic interactions and mechanisms of health-related issues among people experiencing homelessness are still limited. Future studies should focus on the interactions and mechanisms between homelessness and health, as well as barriers and facilitators to utilise healthcare services.

Supplemental Information

Supplemental Information 1 PRISMA checklist.

Additional Information and Declarations

Competing Interests

Author Contributions

Data Availability

The authors declare that they have no competing interests.

Apichai Wattanapisit conceived and designed the experiments, performed the experiments, analyzed the data, prepared figures and/or tables, authored or reviewed drafts of the article, and approved the final draft.

Pannawat Muttarat conceived and designed the experiments, performed the experiments, analyzed the data, authored or reviewed drafts of the article, and approved the final draft.

Tida Sottiyotin conceived and designed the experiments, performed the experiments, analyzed the data, prepared figures and/or tables, authored or reviewed drafts of the article, and approved the final draft.

Pavarud Puangsri analyzed the data, authored or reviewed drafts of the article, and approved the final draft.

Nadvadee Aungkawattanapong analyzed the data, authored or reviewed drafts of the article, and approved the final draft.

Sanhapan Wattanapisit analyzed the data, authored or reviewed drafts of the article, and approved the final draft.

Manas Kotepui analyzed the data, authored or reviewed drafts of the article, and approved the final draft.

The following information was supplied regarding data availability:

This is a literature review and did not generate raw data. The information gleaned from these investigations was synthesized and presented in the Results section.

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
