# Peer review of "Health-related issues of people experiencing homelessness in Thailand: a scoping review"

_PeerJ, doi:10.7717/peerj.17884_

## Round 0.1 · original submission · Major Revisions

Authors are requested to carefully read all reviewers comments and address them accordingly. This manuscript needs major changes.

·

Basic reporting

This study is important in that it has the potential to give a unique overview of this problem in Thailand. The review uses a sound methodology and is well referenced and has a clear Table of results. However, there is a lack of any evaluation of the quality of the papers and some significant grammatical errors that limit its potential to be published in its current form.The results are presented clearly, however further development of the findings is required to make it more unique and relevant to the journal's audience.
The discussion around interactions and mechanisms for health and homelessness could be explored more deeply from the large body of work internationally that has addressed this issue. Many describe the evidence of the number of traumatic events in childhood affecting both homelessness and health.


Some more specific suggestions are detailed below.

The abstract should not get repeated verbatim in the Introduction.

Lines 56-60. It is not necessary to report all the numbers in the introduction.
Line 60 An upward trend has been observed during the pandemic of coronavirus disease 2019 (COVID-19) – Needs a reference.
Methods - Dates for searching.
“From inception” - range of starting dates could be used.

There are a number of sentences with unusual wording and phrases are used inappropriately.

For example -
Line 90 “Thus far”- is not required in the sentence that continues.
.. “it is known that people experiencing homelessness have higher risks of developing several health problems, compared with other populations”
Line 71 “Coincidence” is not correct meaning


There are multiple sentences that need to be edited and do not make sense in their current form - I suggest the entire paper be reviewed by an suitably skilled editor.
For example
Line 162 “A high rate of depression (78.3%) and mental stress (75%) were addressed”
Line 165 “Suicidal risks and alcohol dependence were reported approximately 30% and 20%, respectively”
Line 168 “Severe psychiatric disorders associated with higher risks of suicide and multiple psychiatric disorders”
Line 197 “Family problems (breakdown and poverty), substance use, and attachment to their peers were reported as reasons of children to leave their home”
These sentences do not make sense in their current form.


Grammatical errors are also seen in the following sentence.
Additionally, people with homelessness who drank alcohol in public spaces may lead to alcohol related problems such as physical assaults.
What evidence is there of this causal link?

There are sentences where the meaning in not clear. For example “This may be interpreted as the differences in the Thai context, comparing to other countries or the gap in knowledge of these aspects “

The Table is very detailed and interesting.
Consistent in Table 1. Eg Study Design “ Qualitative study” in some when focus group and interviews are used in others but not described as such. Try and be consistent in the way papers are described.
Capital Letters in Table

“However, studies in Thailand did not focus on the spectrum of healthcare services, including health promotion, disease prevention, treatment, rehabilitation, and palliation” Could you present an opinion as to why this does not occur?


Perhaps some more developed discussion about STIs in this population in other countries or about these problems in Thailand as there seems to be a number of papers on this in the Thai literature?

A very significant deficit of the paper is that there is no comment on the quality of the studies. This may need a separate table and use of a tool to assess the quality of the papers.
There was an emphasis on reports from people that looked after homeless people- not with PEH themselves? This is a real limitation of the studies and creates bias that should be acknowledged. This is relevant if the commentary around the cause and effect.
Line 392 – Should read ”the Proposed…”

Experimental design

See above

Validity of the findings

See above

Reviewer 2 ·

Basic reporting

• Sentence structure isn’t always coherent. Tense in some sections is inconsistent.
• Intro and background offer some context, Thailand section provides good understanding of Thai context, however the global data section could be summarised better. More context about health related impacts of homelessness in recent literature would provide a good starting point.
• Literature is referenced well, some articles are 15-20 years old which needs an explanation of whether these articles are relevant to this paper? This paper would benefit from more context literature about health and homelessness.
• Structure is good.
• The review is broad and cross-disciplinary interest and within the scope of the journal.
• The health-related issues linked to homelessness has been reviewed recently, but not in Thailand and so this article fills the geographical need to understand the Thai context.
• The aims and research question are absent from this review. These are necessary for transparent research reporting.

Experimental design

• The content and topic explored in this article is relevant to the Aims and Scope of PeerJ and sits within the Health Sciences strand of PeerJ.
• No ethical issues as this is literature work. Unfortunately, the methods and process of analysis is not clear and so this cannot be assessed. Themes are referred to, please explain how these themes were identified (I would discourage using the word emerged as I assume your research team created/developed/generated these themes). They read more as topic headings than themes.
• Research methodology, question, aims not defined. Search strategy well explained.
• The research paradigm and values are also not defined, although arguably not necessary, they would add to the strength of this paper.
• Sources are well referenced.
• I would suggest reconsidering the ‘themes’ identified. The named themes read more as topic headings, and this analysis reads more as description of some results rather than an analysis of the body of literature.
• I would also argue that some of the ‘health-related issues’ are more complex than that. For example, prostitution, neglect, sexual abuse and poverty may impact upon health but I would suggest these are social behaviours and experiences linked to homelessness.
E.g.
o Prostition: An high-risk behaviour that may impact upon health/vulnerability.
o Neglect: A social impact (perhaps reference ACEs)
o Sexual abuse: Higher risk of abuse? – yes could be health-related
o Poverty: A socio-economic state of being that again may be linked to poor health but is a stretch to say these are all health-related. There is more to be said here that is not contained in ‘health related issues’.

Validity of the findings

• The argument is not well presented and the conclusion does not summarise the findings of the review.
• There is potential for this article to be impactful, but unfortunately, it is not yet at the level where the conclusions are trustworthy.
• The conclusion suggests two gaps in knowledge. These gaps are unclear throughout the rest of the review and as there is no research question for the literature review, it is unclear how this conclusion can be drawn.

Additional comments

• The topic is important and there is potential here for a strong article and a strong argument for future research.
• However, the argument is not fully formed, and at times is confusing and challenging to read- I would recommend a closer edit of sentence structure, phrasing, tenses and meanings of certain words. I will attempt to attach the document with specific highlighted points.
• The introduction needs to set the scene more clearly, instead of simply providing numbers from other countries, tell a story of how homelessness looks globally, and then how it looks closer to Thailand and then how it looks in Thailand, good use of recent data for this.
• You need to share your research question, aims etc so that the reader has a focus of what is being presented in this article.
• Data has been extracted into a table, but themes are presented- please explain how you generated these themes.
• The themes are topic headings that do not clearly link to the content beneath, perhaps consider changing the names of the themes or reorganising the content.
• The conclusion should link back to your research question, state the answer you got from this review (i.e. what is already known in this area, and what has been excluded from current research) then highlight these gaps and how they may be addressed in future.

Annotated reviews are not available for download in order to protect the identity of reviewers who chose to remain anonymous.

---

## Round 0.2 · accepted · Accept

The authors have addressed the reviewers' comments

·

Basic reporting

Thankyou for addressing these issues. I believe the paper is suitable for publication and will make a valued contribution to the literature

Experimental design

Scoping review

Validity of the findings

Valid

Reviewer 2 ·

Basic reporting

Much improved report, thank you for your work on this.

Sentence Structure:
Very few sentences still need editing, I describe in CAPITALS the changes needed.

Page 11 First sentence under topic 2,
Social behaviours and experiences ARE linked to homelessness.

Compared to youth who live with their families, youth who left homes were more likely to EXPERIENCE child neglect, sexual abuse, rejection, poverty, and being in custody.

Another study found that children and youth experiencing homelessness were more likely to BE LIVING IN family poverty and LACK A close-knit family

Literature:
There is one article that I consider to be out of date for the point made- In the introduction, the first sentence of paragraph 2, homeless populations being at higher risk of developing severe health problems, 1988 - this is 36 years old and there is much more recent research showing the link between health and homelessness

Experimental design

Study design is now clearly explained and described. Thank you.

Validity of the findings

Findings are described clearly with gaps identified, thank you.

Additional comments

Thank you for the changes made, this now reads very well.